# Calibration of Near Infrared Spectroscopy of Apples with Different Fruit Sizes to Improve Soluble Solids Content Model Performance

**DOI:** 10.3390/foods11131923

**Published:** 2022-06-28

**Authors:** Xiaogang Jiang, Mingwang Zhu, Jinliang Yao, Yuxiang Zhang, Yande Liu

**Affiliations:** 1School of Mechatronics & Vehicle Engineering, East China Jiaotong University, Nanchang 330013, China; jxg_ecjtu@163.com (X.J.); zmw15256864150@163.com (M.Z.); a15755567391@163.com (J.Y.); 2Institute of Intelligent Mechanical and Electrical Equipment Innovation, Nanchang 330013, China; 3Huaiyin College of Technology, Huaiyin 223003, China; yxzhang@hyit.edu.cn

**Keywords:** apple, NIR, size correction, extinction coefficient, fruit diameter difference

## Abstract

The transmission spectrum of apples is affected by the fruit’s size, which leads to poor prediction performance of the soluble solids content (SSC) models built for their different apple sizes. In this paper, three sets of near infrared (NIR) spectra of apples with various apple diameters were collected by applying NIR spectroscopy detection equipment to compare the spectra differences among various apple diameter groups. The NIR spectra of apples were corrected by studying the extinction rates within different apples. The corrected spectra were used to develop a partial least squares prediction model for their soluble solids content. Compared with the prediction model of the soluble solids content of apples without size correction, the R_p_ of PLSR improved from 0.769 to 0.869 and RMSEP declined from 0.990 to 0.721 in the small fruit diameter group; the R_p_ of PLSR improved from 0.787 to 0.932 and RMSEP declined from 0.878 to 0.531 in the large fruit diameter group. The proposed apple spectra correction method is effective and can be used to reduce the influence of sample diameter on NIR spectra.

## 1. Introduction

Apples are known for their texture, flavor, visual effect, and nutritional value [1,2]. NIR spectroscopy has become the representative and main development direction of modern non-destructive testing with its unique advantages of simplicity, efficiency, and non-destructiveness, and is an effective way to solve the classification of agricultural products [3,4,5,6]. The application of NIR spectroscopy in fruit and vegetable quality inspection has been reported, mainly focusing on citrus, apple, pear, tomato, and other species [7,8,9,10].

Many scholars have studied the application of NIR spectroscopy in the internal quality of fruits. In terms of algorithms, Travers et al. [11] developed partial least squares (PLS) models based on spectra in the wavelength ranges of 680–1000 nm and 1100–2350 nm, respectively, after extracting the characteristic wavelengths using the competitive adaptive re-weighted sampling algorithm (CARS) for predicting the dry matter (DM) and SSC of pears. The feature wavelengths selected by CARS successfully highlighted the differences between the prediction models based on the two different spectral ranges. In near infrared spectroscopy research, the relationship between near-infrared reflectance and transmittance spectra of kiwifruit and soluble solids was investigated by Schaare et al. [12]. The analysis showed that modeling using transmittance spectra was better than reflectance, with a test set correlation coefficient of 0.961 and a test set root mean square error of 0.8%. Tian et al. [13] used NIR spectra to predict nuclear mold in apples of different fruit sizes, correcting the NIR spectra of apples with different degrees of disease. The accuracy of apple disease degree prediction established by the corrected spectra was up to 90%. In terms of actual testing, Arana et al. [14] examined the soluble solids of white grapes and the method chosen was NIR spectroscopy, which achieved a good prediction of the soluble solids of white grapes. Jha et al. [15] examined the internal quality of seven Indian mangoes by NIR spectroscopy between 1200 nm and 2200 nm for both soluble solids and acidity and established PLS with R_p_ of only 0.715 and 0.703. Liu et al. [16] developed a generalized UVE-PLS model for apple brix by collecting diffuse transmission spectra of red Fuji apples from three different locations, namely Qixia, Luochuan, and Huining, highlighting the potential of spectroscopic techniques for fruit quality detection in different origins. Antonucci et al. [17] conducted a study on the internal quality of oranges by spectroscopic techniques and achieved good results in predicting their acidity and soluble solids by regression analysis using the PLS model, with correlation coefficients of 0.843 and 0.812 for soluble solids and acidity of oranges, respectively. Ni et al. [18] performed the NIR spectral model transfer of different instruments by filtering the wavelength information of different NIR instruments. Two datasets of maize and scutellaria samples measured by different NIR instruments were used to test the performance of the method, where the overall prediction performance of the SWCSS-PLS model for the secondary measurement samples was much better than that of the full-wavelength PLS model. Meng Qinglong et al. [19] collected the reflection spectra of fresh “Fuji” apples from 400 to 1000 nm, and used different pretreatment and different characteristic wavelength screening methods to establish various models to predict the SSC content of apples. It can be better used for the detection of apple SSC. The above studies did not consider the sample size. Ideally, the samples for NIR modeling should include all of the variables affecting the NIR spectra, but this is very difficult for the detection of complex variables in the internal quality of apples. If more variability samples are included in the model, the prediction accuracy of the model decreases and further confirmation is needed to meet the requirements. In this paper, it was found that the light intensity of the transmittance spectra of the internal pulp of apples showed a log-linear relationship with their fruit diameter. Therefore, a size-correction method for apples is proposed, based on which the NIR spectra of all apples with different fruit diameters are transformed into a single spectrum that is used to eliminate the effect of size on the performance of the apple SSC prediction model. Compared with the apple soluble solids content prediction model without size correction, the proposed apple size correction method effectively solves the problem of poor prediction accuracy of apple SSC model due to apple size.

## 2. Materials and Methods

### 2.1. Test Materials

Apples were harvested from a red Fuji apple orchard with 480 apples divided into three fruit size groups (65–75 mm, 75–85 mm, and 85–95 mm), with a total of 160 values under every fruit size set. Spectra information and SSC were collected at the markers.

### 2.2. Spectral Acquisition

The spectra of apples were collected by near-infrared online inspection equipment [20], and the structure of the equipment is shown in Figure 1. The spectrometer was a high-precision spectrometer (QE65Pro, Ocean, Manhattan, NY, USA). The light source system uses 100 W Osram halogen lamps evenly distributed along both sides of the main drive chain, with five lamps on each side. To achieve a stable value of light source attenuation, a white Teflon sphere was used as a reference to calibrate the NIR spectroscopy online detection device after 30 min of warm-up each time the power was turned on. The current value of the regulated power supply was adjusted and the range of energy spectrum change of the transmission spectrum was observed until the standard deviation of the adjacent energy spectrum intensity was within 1% and the the NIR spectra are reproducible, and the apple sample spectrum acquisition was started. The parameters of the spectrum acquisition were set as follows: duration time of 100 ms, motion speed of 5 m/s, and spectral wavelength range of 370–1150 nm.

### 2.3. SSC Measurement of Samples

After collecting NIR spectra of apple samples, 1-cm-thick slices were cut along the equator of the apples and subsequently divided into 4 equal portions according to the label. The SSC of the extracted apple juice was determined with a saccharimeter (PR-101a, ATAGO, Nagasaki, Japan). The measurements were repeated three times to take the average value as the final SSC value.

### 2.4. Data Processing

The 160 sample spectra under each fruit diameter group were divided into a calibration set (120) and a prediction set (40) using the Kennard-Stone (K-S) algorithm. Since the NIR spectral data matrix of each fruit diameter group is 160 × 1044, to reduce the errors caused by non-experimental factors, this study used Unscrambler (Version 9.7, CAMO, City of Oslo, Norway) software to process the spectra using different pretreatment methods (Multiple scattering correction, MSC; standard normal variable transformation, SNV; Savitzky-Golay smoothing, S-G smothing). The partial least squares (PLS) method was then used to establish the apple SSC detection model.

Partial least squares regression (PLSR) is widely used in NIR spectral analysis to decompose the spectral array X and the concentration array Y simultaneously to strengthen the corresponding computational relationship and ensure the best model is obtained. The PLS regression model is shown in Equation (1):(1)Y=bX+e
where *b* denotes the vector of regression coefficients and *e* denotes the model residuals.

The performance of the model is judged by the correlation coefficient R_p_ and the root mean square error value (RMSEP). Equations of R_p_ can be found in Equation (2) and RMSEP can be found in Equation (3).
(2)$$R_{p}=\sqrt{1-\frac{\sum_{i=1}^{n}\left(y_{i}-{\hat{y}}_{i}\right)}{\sum_{i=1}^{n}\left(y_{i}-\bar{y}\right)}}$$
(3)$$\mathrm{RMSEP}=\sqrt{\frac{1}{n-1}\sum_{i=1}^{n}{\left(y_{i}-{\hat{y}}_{i}\right)}^{2}}$$
where *n* is the number of experimental samples, *y_i_* is the actual value of the *i*-th sample in the prediction set measured by the standard method, ${\hat{y}}_{i}$ is the predicted value of the *i*-th sample in the prediction set measured by NIR spectroscopy and mathematical model, and ${\bar{y}}_{i}$ is the mean value of the SSC of all apples in the prediction set.

## 3. Results and Analysis

### 3.1. Sample Chemical Index Statistics Results

The 160 apple spectra under the calibration set (120) and the prediction set (40) by the K-S algorithm were sorted and the sorted apple SSC values are presented in Table 1. The SSC range of the modeling set under each fruit size group was larger than the SSC range of the prediction set, which allows for improved forecasting of apple SSC.

### 3.2. Near-Infrared Spectra of Three Groups of Fruit Size Apples

The mean spectra of apples under each fruit size group are shown in Figure 2. With the increase of the fruit diameter, the corresponding spectra energy of apples is smaller. The strongest spectra energy was collected from 65–75 mm apples, and the weakest spectra energy was collected from 85–95 mm apples. The effective wavelength range was set from 350 to 850 nm due to the weak signal and little effective information at both ends of the spectra. The spectral trends of apples with different fruit sizes were the same, with differences in absorption intensity. The spectral curves showed prominent absorption peaks near 645, 710, and 810 nm, and troughs near 675, 758, and 830 nm, respectively. The absorption peak at 645 nm was mainly influenced by the color of the epidermis [21], the absorption of chlorophyll near 675 nm might be the absorption of the chlorophyll [22], and the trough near 758 nm was related to the O-H triplet stretching vibration [23], and the weaker trough near 830 nm was related to the N-H triplet stretching vibration [24].

As can be seen in Figure 2, the intensity of the NIR spectra is decreasing as the fruit diameter increases. This is due to the fact that there is an attenuation of the NIR light intensity due to the flesh of the fruit when the light is transmitted inside the apple. The degree of attenuation of the NIR light intensity increases with the increase of the light range. The degree of attenuation of NIR light intensity inside the apple shows a logarithmic relationship with the apple fruit diameter [25], which can be fitted as a function of Equation (4).
(4)$$I=I_{0}{\exp(-u}_{e}d)$$
where *I_0_* is the light intensity emitted by the NIR source, *I* is the received NIR spectral intensity, *d* is the light transmission length, and *u_e_* is the attenuation degree factor of NIR spectra.

In the spectral acquisition device shown in Figure 1, *d* is the fruit diameter of the apple. As *d* increases, the NIR light intensity becomes more attenuated during the propagation inside the apple, and the intensity of the obtained apple NIR spectra becomes smaller. It can be seen that the difference in apple fruit diameter will affect the light intensity of its NIR spectra. 

### 3.3. PLSR Results of SSC for Mixed Apple Size 

To verify that apple size differences affect their NIR spectra and lead to poor prediction performance of the developed apple SSC prediction model. The calibration and prediction sets of fruit diameter groups 65–75 mm, 75–85 mm, and 85–95 mm were used as the calibration and prediction sets of the mixed fruit diameter apple prediction model to establish the PLSR of SSC for different apple sizes, and the model effects are shown in Table 2.

As can be seen from Table 2, the SSC model built with mixed apple fruit diameter has a poor prediction performance with an R_p_ of 0.722 and an RMSEP of 1.086. When there is a large difference in apple fruit diameter in the model, it will cause the problem of poor prediction performance of the established model, so size correction of apple fruit diameter is needed to improve the prediction performance of its SSC model.

### 3.4. PLSR Results of SSC for Each Fruit Size Set 

The number of LVs in the PLSR model was set from 1 to 20 to prevent the overfitting or underfitting of the model. Table 3 shows the PLSR results established for individual fruit size sets after several pretreatment methods.

From Table 3, it can be seen that the PLSR prediction performance of the SSC established by SNV pretreatment of apple NIR spectra for the three sets of fruit sizes is the best, and the correlation coefficients R_p_ of the models are 0.863, 0.947, and 0.917, respectively, and the root mean square error values RMSEP of the prediction sets are 0.771, 0.622, and 0.752, respectively. The scatter plot of PLSR prediction is shown in Figure 3. It can be seen that the pretreatment method SNV can eliminate the effect of sample particle size on NIR spectra [26], thus solving the influence of spectral dispersion due to the unequal sample dimensions.

### 3.5. Individual Fruit Size Groups Predicted Other Fruit Size Groups

It can be seen from Table 3, the PLSR prediction performance of SSC established when the apple size was 75–85 mm was better. To investigate whether the prediction performance of the model could be achieved when more variance samples were included in the model, the PLSR results for the remaining two fruit size groups using the medium apple size group to predict the SSC are shown in Table 4.

As can be seen from Table 4, the PLSR prediction performance of soluble solids content built from the modeling and prediction sets was poor when the difference in apple fruit size was significant between them. Compared with Table 3, the correlation coefficient R_p_ decreased from 0.863 to 0.769 and the root means square error value RMSEP increased from 0.771 to 0.990 for the PLSR of the small fruit size group. The correlation coefficient R_p_ decreased from 0.917 to 0.787 and the root means square error value RMSEP increased from 0.752 to 878 for the PLSR of the large fruit size group. The scatter plot of its PLSR is shown in Figure 4. Apple size differences significantly impacted the accuracy of the SSC model. With the same variety of apples, there will be differences in volume size, and the size differences will affect the detection performance when performing NIR spectroscopy, so it is necessary to correct the NIR spectra of apples of different sizes to improve the detection performance of NIR detection equipment.

### 3.6. Correction of Near Infrared Spectroscopy for Apples of Various Diameters

From Formula (4), apple size affects the prediction model of soluble solids content, and the light intensity of the apple and its fruit diameter are logarithmic functions. The deformation of Formula (4) can be obtained as Formula (5).
(5)$$\ln(I)=\ln(I_{0})-u_{e}d$$

If the light intensity of the apple at its two internal depths *d_1_* and *d_2_* are *I_1_* and *I_2_*, respectively, Formula (5) can be deformed as:(6)$$-u_{e}=\frac{\ln(I_{1})-\ln(I_{2})}{d_{1}-d_{2}}$$

From Formula (5), we can find the extinction coefficient of apples or the collection method in Figure 1, the light range *d* is the fruit diameter at the equator of apples, and *I* is the light intensity of the transmission spectra of apples collected by the fiber optic probe. From Table 3, it can be seen that the PLSR performance of SSC with medium apple size is better, so the average spectra of the medium fruit size group are taken as IR and the average fruit size of the medium fruit size group is taken as *d_1_*, and the average extinction coefficients of all samples can be obtained as shown in the following Formula (7).
(7)$$-u_{e}=\frac{\sum_{i=1}^{n}\left(\frac{\ln\left(I_{R}\right)-\ln\left(I_{i}\right)}{d_{R}-d_{i}}\right)}{n}$$
where *I_R_* is the reference spectra, *I_i_* is the spectra of the apple sample *i*-th, *d_R_* is the reference fruit diameter, *d_i_* is the average fruit size of the apple sample *i*, *n* is the number of samples, and the extinction coefficient applicable to all apple samples can be obtained from Formula (7). The inverse operation of Formula (7) leads to Formula (8).
(8)$$I_{i}^{*}=\exp\left(-u_{e}\left(d_{R}-d_{i}\right)+\ln\left(I_{i}\right)\right)$$
where *I_i_^*^* is the size-corrected spectra of apple sample *i* according to its fruit diameter. The size-corrected spectra of all samples can be obtained according to Formula (8), and their average size-corrected spectra of different apple fruit diameter groups are shown in Figure 5.

Compared with the uncorrected apple spectra, the size-corrected apple spectra have a cross-over phenomenon in the spectra of each fruit diameter group, and the spacing between the vertical aspects of each fruit diameter group in the spectra is reduced compared with Figure 2. This spacing exists as a result of the differences in apple fruit diameter. Our proposed size correction approach was used to correct the near-infrared spectra for apples of various dimensions.

The corrected NIR spectra of apples were used to build the PLSR of SSC with different fruit sizes. The large and small fruit size groups were predicted using the medium fruit size group, and the predicted results are shown in Table 5.

As can be seen from Table 4 and Table 5, the model prediction performance of the corrected NIR spectra compared to the PLSR built for the original apple spectra was significantly improved. Among them, the correlation coefficient R_p_ of the PLSR established for the small fruit size group improved from 0.769 to 0.869, and RMSEP decreased from 0.990 to 0.721. the correlation coefficient R_p_ of the PLSR established for the large fruit size group improved from 0.787 to 0.932. the RMSEP decreased from 0.878 to 0.531. the PLSR of the two fruit size groups scatters plots are shown in Figure 6. The results show that after the spectral correction of Formula (7), the spectra of apples of different sizes can be converted into a standard spectrum to correct the NIR spectra of apples of different sizes, which is used to improve the performance of its prediction model. Similarly, this spectral correction method can be applied to other fruits such as pear, citrus, and watermelon.

## 4. Conclusions

This paper presents a method that can correct the near-infrared spectra of apples with various dimensions to enhance the performance of SSC prediction models. The transmission spectra of apples of different sizes were converted to the same standard to obtain the extinction coefficients of transmitted light. The transmission spectra were corrected according to the average extinction coefficients of apples, and the corrected spectra were employed to model the soluble solids content of apples. Compared with the model of the soluble solids content of apples without size correction, the correlation coefficient R_p_ of PLSR for the small fruit diameter group increased from 0.769 to 0.869 and RMSEP decreased from 0.990 to 0.721. The correlation coefficient R_p_ of PLSR for the large fruit size group increased from 0.787 to 0.932 and RMSEP decreased from 0.878 to 0.531. The apple size correction method proposed in this paper is reliable. By improving the model algorithm in the application of NIR online inspection device, the standard spectra are set to size correct the apple spectra of different fruit diameters and reduce the influence of apple fruit size variation on its SSC model.

## Figures and Tables

**Figure 1 foods-11-01923-f001:**
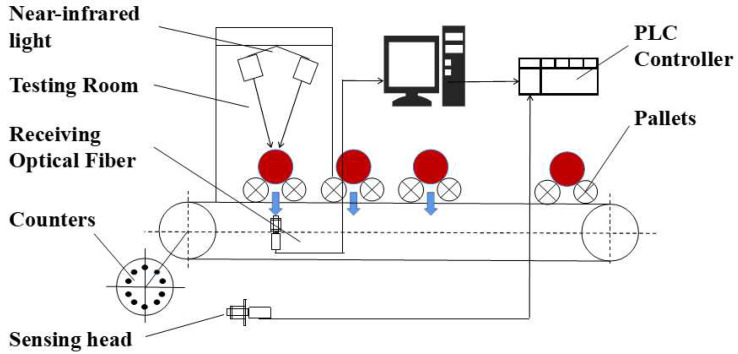
Near-infrared spectra acquisition device.

**Figure 2 foods-11-01923-f002:**
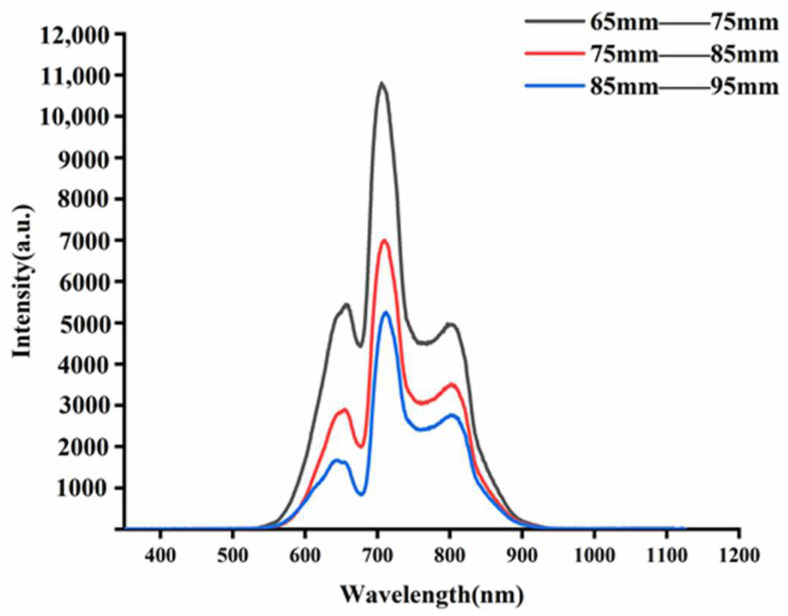
Raw spectra of three sets of fruit diameter samples.

**Figure 3 foods-11-01923-f003:**
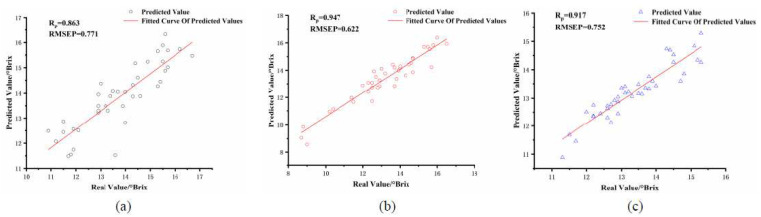
Scatter plot of PLSR prediction of apple SSC for three apple diameter sets. (**a**) apple size 65–75 mm, (**b**) apple size 75–85 mm, and (**c**) apple size 85–95 mm.

**Figure 4 foods-11-01923-f004:**
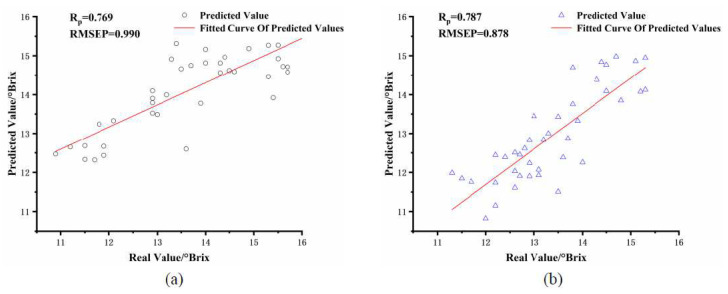
Scatter plot of PLSR with different fruit diameter of modeling set and prediction set. (**a**) 75–85 mm apple size group predicted 65–75 mm apple size group, (**b**) 75–85 mm apple size group predicted 85–95 mm apple size group.

**Figure 5 foods-11-01923-f005:**
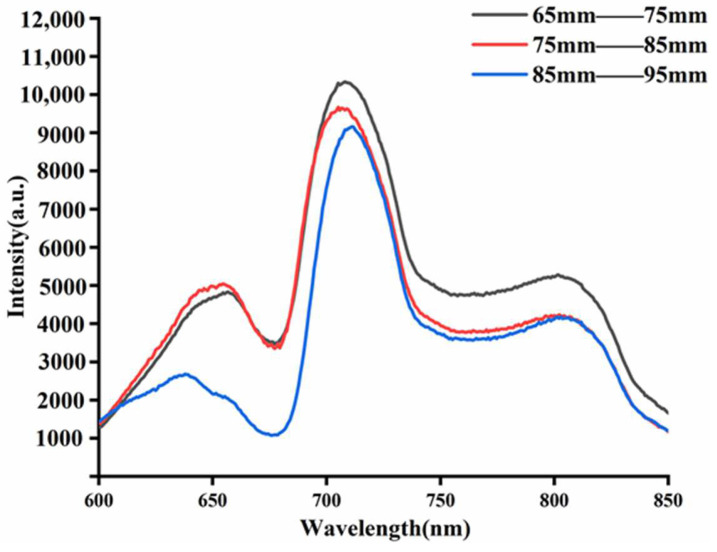
Apple spectra after size correction.

**Figure 6 foods-11-01923-f006:**
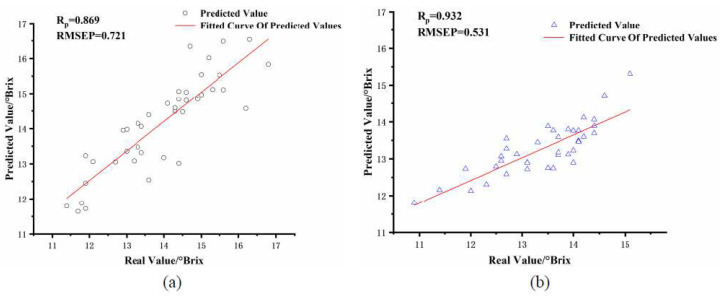
The PLSR scatter plot was created by the corrected spectra. (**a**) 75–85 mm fruit size group predicted 65–75 mm fruit size group, (**b**) 75–85 mm fruit size group predicted 85–95 mm fruit size group.

**Table 1 foods-11-01923-t001:** SSC values for apples of different fruit sizes.

Fruit Size Range	Data Type	Number of Samples	Max/Brix	Min/Brix	Mean/Brix	Deviation
65–75 mm	Calibration Set	120	17.2	9.8	13.8	1.62
	Prediction set	40	16.7	10.9	13.8	1.24
75–85 mm	Calibration Set	120	17.2	8.3	13.2	1.60
	Prediction set	40	16.5	8.7	12.73	1.78
85–95 mm	Calibration Set	120	15.3	10.9	13.3	1.73
	Prediction set	40	15.3	11.3	13.3	1.92

**Table 2 foods-11-01923-t002:** PLSR findings for SSC of mixed apple size groups.

Number of Calibration Set	Number of Prediction Set	R_c_	RMSEC	R_p_	RMSEP
360	120	0.733	1.011	0.722	1.086

**Table 3 foods-11-01923-t003:** PLSR findings for SSC of various apple size groups after spectra pretreatment.

Fruit Size Range	Pretreatment	LVs	R_c_	RMSEC	R_p_	RMSEP
65–75 mm	Original	11	0.931	0.592	0.853	0.786
	MSC	8	0.902	0.700	0.857	0.785
	SNV	12	0.972	0.376	0.863	0.771
	S-G smoothing	12	0.907	0.683	0.854	0.794
75–85 mm	Original	10	0.951	0.534	0.941	0.654
	MSC	9	0.964	0.462	0.941	0.654
	SNV	11	0.976	0.373	0.947	0.622
	S-G smoothing	11	0.950	0.540	0.937	0.677
85–95 mm	Original	10	0.916	0.389	0.898	0.827
	MSC	9	0.908	0.371	0.814	0.817
	SNV	10	0.936	0.295	0.917	0.752
	S-G smoothing	11	0.909	0.369	0.854	0.852

MSC: multivariate scattering correction; SNV: standard normal variables transformation; S-G smoothing: Savitzky-Golay smoothing; Lvs, latent variable individual.

**Table 4 foods-11-01923-t004:** The fruit size group alone predicted the SSC results of other fruit size groups.

Calibration Set	Prediction Set	R_c_	RMSEC	R_p_	RMSEP
75–85 mm	65–75 mm	0.951	0.534	0.769	0.990
	85–95 mm	0.958	0.412	0.787	0.878

**Table 5 foods-11-01923-t005:** PLSR results were established after the spectra correction.

Calibration Set	Prediction Set	R_c_	RMSEC	R_p_	RMSEP
75–85 mm	65–75 mm	0.951	0.570	0.869	0.721
	85–95 mm	0.969	0.459	0.932	0.531

## Data Availability

The data presented in this study are available on request from the corresponding author.

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
