# Peer review of "Calibration of Near Infrared Spectroscopy of Apples with Different Fruit Sizes to Improve Soluble Solids Content Model Performance"

_foods, 2022, doi:10.3390/foods11131923_

Round 1

Reviewer 1 Report

The manuscript presents new information on the quantification of soluble solids content in apples, based on statistical analyzes of various properties of the samples. The authors present theoretical and experimental rigor, with a deep discussion of the results. The manuscript is interesting and need major revision. My biggest concern goes to the lack of important information in the Materials and methods section. The following are some suggestions/corrections that can help in better understanding and reproducing this work:

Introduction section: the authors should emphasize the novelty of the research

Line 75. Why did the authors present their results in the introduction section?

Line 104 were NIR spectra measured with replications?

Line 116. Please define the version and producer of the used software

In the Data processing section please describe how big was data matrix? Were the NIR spectra pre-processed? What pre-processing methods were used? How were data divided into validation and prediction sets?

Table 1. I suggest presenting the SSC values as the average value ± st. deviation

Applicability of the developed PLS models should be estimated based on: (i) the coefficients of determination for calibration (Rcal2) and cross-validation (Rcval2), (ii) root mean square error for calibration (RMSEC) and cross-validation (RMSECV), (iii) standard error for calibration (SEC) and cross-validation (SECV), (iv) average value of the difference between predicted and measured values (Bias), (v) ratio of predicted deviation (RPD) and range error ratio (RER)

Author Response

Dear editors and reviewers.

Thank you for your letter and for the reviewers' comments on our manuscript entitled ‘Calibration of Near Infrared Spectroscopy of apples with different fruit sizes to improve soluble solids content model performance’. These comments were very valuable and helpful. We have read the comments carefully and have made corrections. In accordance with the instructions in your letter, we have uploaded the file of the revised manuscript. Here are our responses to the reviewers' comments:

Response to Reviewer Comments

Reviewer 1:

  1. Introduction section: the authors should emphasize the novelty of the research.

Response: A description of the novelty of the study has been added to the introduction, lines 77 to 87 in the text.

  1. Line 75. Why did the authors present their results in the introduction section?

Response: The results of the previous study were added to show that the previous study did not take the sample size into account, so the predictive performance of the model built was average, thus proving that a correction for the size of the sample was necessary.

  1. Line 104 were NIR spectra measured with replications?

Response: It is the NIR spectra, and for the NIR online detection device, the intensity of the internal light source or the transmission speed will affect its NIR spectra, so the instrument should be calibrated before collecting the NIR spectra of apples. It has been explained in the corresponding section of the text, lines 100 to 110.

  1. Line 116. Please define the version and producer of the used software

Response: The versions and manufacturers of the software used in the article have been described, line 122.

  1. In the Data processing section please describe how big was data matrix? Were the NIR spectra pre-processed? What pre-processing methods were used? How were data divided into validation and prediction sets?

Response: The NIR spectra matrix of each fruit diameter group is 160 × 1044, the spectra are preprocessed by MSC, SNV and S-G smoothing, the data are divided between validation and prediction sets by the K-S algorithm, which has been modified in the data processing section of the paper, lines 120 to 130.

  1. Table 1. I suggest presenting the SSC values as the average value ± st. deviation

Response: The deviation of the Apple SSC has been added to Table I in reference to the reviewer's suggestion, line 156.

  1. Applicability of the developed PLS models should be estimated based on: (i) the coefficients of determination for calibration (Rcal2) and cross-validation (Rcval2), (ii) root mean square error for calibration (RMSEC) and cross-validation (RMSECV), (iii) standard error for calibration (SEC) and cross-validation (SECV), (iv) average value of the difference between predicted and measured values (Bias), (v) ratio of predicted deviation (RPD) and range error ratio (RER).

Response: The evaluation of PLS model applicability in this paper uses (ii) RMSEC and RMSECV as mentioned by the reviewer, because this paper divides the calibration and validation sets, and in order to see more obviously the effect of apple fruit diameter differences on model prediction performance, the root mean square error value of the prediction set RMSEP is used instead of RMSECV. and both model correlation coefficients Rc and Rp are used to evaluate the model. The performance was evaluated using both model correlation coefficients Rc and Rp, which are described in Section 2.4 of the paper. Rc and RMSEC have been added to the model evaluation metrics according to the reviewers' comments. Line 199; line 225; line 284.

Reviewer 2 Report

This manuscript contains sufficient novelty, but still minor modifications and suggestions are recommended to improve the quality.

  All minor remarks are highlighted in the manuscript.

·       Authors should avoid the lumping of references in the paper, but each should be discussed.

·       References should be arranged according to the instructions for the authors.

Author Response

Dear editors and reviewers.

Thank you for your letter and for the reviewers' comments on our manuscript entitled ‘Calibration of Near Infrared Spectroscopy of apples with different fruit sizes to improve soluble solids content model performance’. These comments were very valuable and helpful. We have read the comments carefully and have made corrections. In accordance with the instructions in your letter, we have uploaded the file of the revised manuscript. Here are our responses to the reviewers' comments:

Response to Reviewer Comments

Reviewer 2:

  1. All minor remarks are highlighted in the manuscript.

Response: Revisions have been made in accordance with the reviewers' comments.

  1. Authors should avoid the lumping of references in the paper, but each should be discussed.

Response: References in the text have been discussed category by category, and the introductory section has been revised. Line 38-53.

  1. References should be arranged according to the instructions for the authors.

Response: The formatting of the references has been modified in accordance with the author's instructions. Line 330-384.

Reviewer 3 Report

The article describes a method of apple size correction which effectively resolves the problem of low prediction accuracy of apple SSC models due to the size of apples. The purpose and methodology of this study seem to be of interest. 

1- The introduction should be modified to include papers that use spectral correction and model transfer in order to improve performance.

2- What type of validation procedure was used? It should be explained, and the results of the validation and training sets should be provided.

3- Authors need to demonstrate poor performance without regard to sample size before starting modeling in different fruit sizes. Accordingly, models should be provided first that include all fruit sizes. 

4- The paper lacks an extensive discussion of the results obtained and should be thoroughly revised.

5- The practical application of this method must be discussed in the conclusion.

Author Response

Dear editors and reviewers.

Thank you for your letter and for the reviewers' comments on our manuscript entitled ‘Calibration of Near Infrared Spectroscopy of apples with different fruit sizes to improve soluble solids content model performance’. These comments were very valuable and helpful. We have read the comments carefully and have made corrections. In accordance with the instructions in your letter, we have uploaded the file of the revised manuscript. Here are our responses to the reviewers' comments:

Response to Reviewer Comments

Reviewer 3:

  1. The introduction should be modified to include papers that use spectral correction and model transfer in order to improve performance.

Response: References for spectral correction and model transfer have been added to [13] and [18], respectively. Line 52-56; Line 71-78.

  1. What type of validation procedure was used? It should be explained, and the results of the validation and training sets should be provided.

Response: The verification program used in the paper is PLS in Unscrambler software, and the software has been described and modified in Section 2.4 of the paper. The results RMSEC and RMSEP for the validation and training sets have been added. Line 133-159.

  1. Authors need to demonstrate poor performance without regard to sample size before starting modeling in different fruit sizes. Accordingly, models should be provided first that include all fruit sizes.

Response: The apple SSC prediction model using all apple samples without considering sample sizes has been added as requested by the reviewer, section 3.3 of the text, lines 206-221.

  1. The paper lacks an extensive discussion of the results obtained and should be thoroughly revised.

Response: Extensively discussed modifications have been made to all results obtained in the text. Line 264-269; Line 323-239;

  1. The practical application of this method must be discussed in the conclusion.

Response: The practical application of the method has been discussed in the conclusion in accordance with the reviewers' comments. Line 349-353.

Round 2

Reviewer 1 Report

The authors put an effort and answered the comments and suggestions. In my opinion they have improved the manuscript and therefore I think that the manuscript can be accepted for publication.